# Should We Scale-Up? A Mixed Methods Process Evaluation of an Intervention Targeting Sedentary Office Workers Using the RE-AIM QuEST Framework

**DOI:** 10.3390/ijerph17010239

**Published:** 2019-12-29

**Authors:** Bradley MacDonald, Ann-Marie Gibson, Xanne Janssen, Jasmin Hutchinson, Samuel Headley, Tracey Matthews, Alison Kirk

**Affiliations:** 1School of Psychological Sciences and Health, University of Strathclyde, 16 Richmond Street, Glasgow G1 1XQ, UK; annmarie.gibson@strath.ac.uk (A.-M.G.); xanne.janssen@strath.ac.uk (X.J.); alison.kirk@strath.ac.uk (A.K.); 2Department of Exercise Science and Athletic Training, Springfield College, 263 Alden Street, Springfield, MA 01109, USA; jhutchinson@springfieldcollege.edu (J.H.); sheadley@springfieldcollege.edu (S.H.); 3School of Physical Education, Performance and Sport Leadership, Springfield College, 263 Alden Street, Springfield, MA 01109, USA; tmatthews@springfieldcollege.edu

**Keywords:** sedentary, sitting, office workers, workplace health, process evaluation, RE-AIM, scale-up

## Abstract

*Background*: Interventions targeting a reduction in sedentary behaviour in office workers need to be scaled-up to have impact. In this study, the RE-AIM QuEST framework was used to evaluate the potential for further implementation and scale-up of a consultation based workplace intervention which targeted both the reduction, and breaking up of sitting time. *Methods:* To evaluate the Springfield College sedentary behaviour intervention across multiple RE-AIM QuEST indicators; intervention participant, non-participant (employees who did not participate) and key informant (consultation delivery team; members of the research team and stakeholders in workplace health promotion) data were collected using interviews, focus groups and questionnaires. Questionnaires were summarized using descriptive statistics and interviews and focus groups were transcribed verbatim, and thematically analysed. *Results*: Barriers to scale-up were: participant burden of activity monitoring; lack of management support; influence of policy; flexibility (scheduling/locations); time and cost. Facilitators to scale up were: visible leadership; social and cultural changes in the workplace; high acceptability; existing health and wellbeing programmes; culture and philosophy of the participating college. *Conclusions*: There is potential for scale-up, however adaptations will need to be made to address the barriers to scale-up. Future interventions in office workers should evaluate for scalability during the pilot phases of research.

## 1. Introduction

Adult office workers spend as much as 80% of their working day engaging in sedentary behaviour (SB) or sitting [1]. Sitting at work has emerged as a global health issue due to the increasing evidence that sitting time is associated with type 2 diabetes, cardiovascular disease and mortality [2,3,4]. Additionally, there is evidence to suggest that the disease risk is greater if an individual accumulates sitting time in uninterrupted bouts [4,5]. The disease risk associated with sitting carries a considerable societal and economic burden. For example, in the United Kingdom alone, sitting time is conservatively estimated to be associated with over 16,000 preventable deaths a year; costing the country’s health care system an estimated 700 million pounds a year [6].

Reducing the burden of disease is contingent on effective workplace interventions being implemented at scale; and, although interventions have been effective [7,8], a significant proportion have been conducted on a relatively small scale [9]. Furthermore, in these interventions, there is a clear emphasis on reporting indicators of efficacy, and a failure to measure and/or report on indicators that would inform the potential for scale-up and sustainability (e.g., participation rate and/or cost) [9,10,11]. Failure to measure and report on additional indicators relating to the reach, implementation and maintenance of the intervention may mean that barriers to scaling up go unnoticed, making the goal of population-level risk reduction unattainable [9,12]. Collecting data and comprehensively evaluating interventions across additional indicators will help inform the researchers’ understanding of the potential for translation and implementation at scale [9,10,11,12].

The RE-AIM Qualitative Evaluation for Systematic Translation (QuEST) mixed methods evaluation framework can facilitate a comprehensive evaluation of an intervention across five dimensions (R-reach, E-effectiveness/efficacy, A-adoption, I-implementation and M-maintenance) [13]. Reach is defined as the absolute number, proportion, and representativeness of eligible individuals who participate in a given initiative. Effectiveness/efficacy refers to the impact of an intervention on the relevant outcomes, including potential adverse effects, quality of life, and economic outcomes. Adoption looks at the reach and effectiveness/efficacy of an intervention at the setting level. It is defined as the absolute number, proportion, and representativeness of the settings and intervention agents (a group of people who implement the intervention) who are willing to initiate a program. Implementation refers to the intervention agents’ fidelity to the various elements of an intervention’s protocol. This includes consistency of delivery as intended; and the time and cost of the intervention. The maintenance dimension is concerned with both the setting, and the individual level; at the setting level, maintenance is the extent to which a program or policy becomes institutionalised or part of organisational practices and policies; at the individual level, maintenance is considered the monitoring of effectiveness of an intervention or program six months or more after the most recent contact [10,14].

RE-AIM QuEST facilitates a broadened approach by enabling qualitative enquiry to further explore and report on additional indicators which would inform the potential for scale-up across the RE-AIM dimensions [13]. For example, questions such as: What are the barriers and facilitators to each dimension? What are the conditions and mechanisms that lead to effectiveness? What are the contextual factors and processes underlying barriers and facilitators to further implementation [13]? In this study, the RE-AIM QuEST framework was used to evaluate the potential for further implementation and scale-up of a consultation based workplace intervention which targeted both the reduction, and breaking up of sitting time.

## 2. Materials and Methods

### 2.1. Springfield College Sedentary Behavior Intervention

The intervention aimed to explore the effect of a consultation based personalised intervention on both reducing total sitting time and increasing breaks in sitting time among desk based US college employees. Participants were first asked to wear an activPAL accelerometer to objectively measure sitting time over one week. This individual data was analysed and given as feedback as part of a behavioural consultation. The consultation was underpinned by Lewin’s force field theory and sought to increase driving forces for change and reduce restraining forces [15]. The consultation consisted of one 45 min face-to-face session conducted by a member of the consultation team. This was followed by a series of weekly follow-up emails delivered over 16 weeks. 87 employees participated in the consultation intervention and a sub subsample of 36 participants wore an activPAL at follow-up. Changes in objectively measured sitting outcomes have been previously reported in Hutchinson et al. [8]. In brief, 36 participants (seven men, 29 women; mean age, 51.1 ± 11.1 years; mean body mass index (BMI), 29.2 ± 7.6 kg/m^2^) completed data collection and as result of the intervention, the number of prolonged sitting bouts >30 min decreased significantly by 0.52 bouts/day (*p* = 0.010) [8].

### 2.2. Evaluation Participants

Participants included (a) intervention participants, (b) Non-participants (employees who completed the Occupational Sitting and Physical Activity Questionnaire (OSPAQ) prior to the intervention, but decided not to participate in the intervention); and (c) key informants. Key informants included; members of the consultation delivery team; study coordinator; members of the research team and additional stakeholders of the health programs on campus. All evaluation participants provided informed consent.

### 2.3. Process Evaluation Study Design

This evaluation utilises both qualitative and quantitative methods to collect data across four of the RE-AIM dimensions. Information on the “adoption” dimension was not collected as the intervention was implemented in only one setting. Data collection was informed and guided by the RE-AIM QuEST mixed methods framework for program evaluation [13]. This framework has been developed to enhance a RE-AIM evaluation by facilitating both quantitative and qualitative exploration of each of the five RE-AIM dimensions; looking to understand the how and why behind intervention implementation [13].

### 2.4. Data Collection

Data were collected for the RE-AIM evaluation retrospective to the completion of the intervention by a researcher independent of, and based at a different university from, the original research team. Ethical approval for the evaluation of the intervention was granted from both Universities’ ethics committees.

#### 2.4.1. Qualitative Data Collection

##### Interviews and Focus Groups with Intervention Participants

Fifteen participants (14 female and one male) who took part in the intervention were invited, via e-mail, to participate in the evaluation. Each consenting participant took part in one of two focus groups Two participants who could not find a convenient focus group time were offered individual interviews, and both were interviewed individually using the same semi-structured topic guide. The focus groups and Interviews were all conducted at the participants place of work in a meeting room. All were recorded using a Dictaphone (OLYMPUS, Tokyo, Japan). 

The topic guide for participants was developed in line with the RE-AIM QuEST mixed methods framework. The topic guide was developed, piloted and refined prior to the focus groups and interviews taking place. Semi-structured interviews were approximately 40 min in length (*n* = 2) and focus groups were approximately 1 h in length (*n* = 2). Participants of the focus groups and interviews also completed a demographics questionnaire prior to starting the focus group or interview. Some example questions from the interview guide are highlighted below (Table 1).

##### Interviews with Key Informants

Seven key informant interviews were conducted either in person or over the phone. Key informants were identified through the research team and were emailed to be a part of the evaluation. Those who responded were then sent an information sheet, consent form and a convenient time and place were scheduled for the interview. A key informant interview guide was developed in line with the RE-AIM QuEST framework. The key informant interview guide was developed separately to the participant topic guide due to their different experience with the intervention. There was very minimal overlap in questions however the core themes of questions remained the same (see Table 2). The interview guide required some minimal adaptations in order to reflect the specific key informants’ involvement with the intervention. The interviews varied in length (23 min–67 min).

#### 2.4.2. Questionnaire Data Collection

The quantitative data was collected through three individual questionnaires (Appendix A). These three questionnaires were developed specifically for obtaining feedback for this intervention and were given to three unique evaluation participant groups. These groups included; non-participants, participants who dropped out and participants who completed the intervention. Both the non-participants and participants who dropped out were sent links to a brief two-item online (Qualtrics) questionnaire to explore reasons for non-participation and dropout. In the two similar questionnaires, employees were first asked why they did not participate. They were given a list of potential options which slightly differed depending on if they were a non-participant or a dropout participant. The second item in both questionnaires was a free text box in which participants could further explain their answer or give an alternative answer. The third questionnaire was developed for participants who completed the consultation intervention. Each were sent a link to a nine-item post intervention questionnaire in which they answered questions on regarding their participation. For example, participants were asked “Did your awareness of time spent sitting change following the intervention: (a) A lot (b) A fair amount (c) Moderately (d) A little (e) Not at all.

In addition, participation rate was calculated based on information obtained from the college’s department of human resources and the study coordinator regarding employment numbers and the study response numbers. Cost of implementation (in hours) was calculated based on information obtained from the study coordinator regarding the total study hours used for training the consultation team, and the total hours worked by the consultation team delivering consultations to participants.

### 2.5. Measures

Table 3 illustrates each of the indicators assessed in this process evaluation to inform on specific dimensions; and the data source used to measure and/or understand each indicator.

### 2.6. Data Analysis

#### 2.6.1. Qualitative

Braun and Clarke’s [16] approach to thematic analysis was used to separately analyse both the study participant data and the key informant data. This approach was selected for its adaptability to both participant and key informant interview data. Furthermore, Braun and Clarke also advocate for the approach’s theoretical flexibility, which facilitated the use of both inductive open coding across the data, as well as deductive coding, based on the RE-AIM framework [16,17].

Familiarisation—All of the data collection and analysis was done by the lead researcher (B.M.). With a wider understanding of the researcher’s central place in the interpretation of the data [18] the lead researcher listened back to the recordings after completion of the focus groups and interviews and created reflexivity notes [18]. The interviews were then listened to again and transcribed verbatim.

Generation of initial codes—The transcripts were uploaded onto an analysis software tool Nvivo (12) to facilitate organisation of the coding process. The lead researcher performed all of the initial coding by creating initial codes, which pulled together text, that the lead researcher considered analytically important in relation to the research question. Deductive coding was carried out in relation to the RE-AIMQuEST framework, aligning data to one of the five indicators of the framework. Inductive open coding of the data was also carried out to ensure information that did not specifically relate to the indicators of RE-AIM was not lost. To enhance trustworthiness of the data, a second sweep of coding was conducted [17] in which both the lead researcher, and another experienced qualitative researcher or “critical friend” (A.-M.G.), interrogated the initial interpretation [19,20]. This coding process was used firstly for all of the intervention participant data. It was then separately repeated for the key informant data, with the analysis being more deductive in nature. Finally, it was repeated for the free text responses from the questionnaires, with the analysis being purely inductive in nature.

Generation of themes—Similar coding constructs were brought together into initial themes by the lead researcher. At this point, to further enhance trustworthiness of the data, a process of critical examination was employed. The lead researcher and critical friend met on four occasions. In these meetings each initial theme was interrogated by both the critical friend and the lead researcher. Through this process, written feedback was generated for each theme. After reviewing and reflecting on the feedback, the lead researcher revisited the theme constructs, making changes to the initial themes, and subsequently, renamed and defined each theme. Quotes were then selected which best represented the central organising concept within each theme.

#### 2.6.2. Questionnaire Data Analysis

Questionnaire data was analysed in SPSS using descriptive statistics to understand the frequency of responses for each question. The free text responses for all of the questionnaires were uploaded into Nvivo and coded for themes.

## 3. Results

A total of 148 individuals participated in the evaluation. One male and 14 female (aged between 47–64) office based workers who took part in the intervention, and self-reported sitting daily sitting time per working day as >8 h, participated in one of two focus groups (*n* = 5, *n* = 8) or an individual interview (*n* = 2). Additionally, seven interviews were carried out with members of the consultation team (*n* = 3), study coordinator (*n* = 1) and additional stakeholders of workplace health (*n* = 3). Sixty-nine office-based employees completed the non-participant questionnaire; seven employees completed the drop-out questionnaire, and sixty-one employees completed the 9-item post intervention questionnaire. The results are presented within the dimensions of the RE-AIM framework to clearly illustrate where data relates to individual dimensions.

### 3.1. Reach

#### 3.1.1. Participation Rate

Of the 680 university employees, 376 completed the baseline OSPAQ questionnaire (55%). All of the 376 employees were then asked to participate in the study, and 87 participants enrolled; equalling approximately 15% of the original eligible employee population. The questionnaire data and four qualitative themes outlined below highlight the facilitators and barriers to high participation, reported by evaluation participants.

#### 3.1.2. Facilitators of Enrolment to the Intervention

Theme 1—Inclusive participation and feeling welcome

Participants expressed that the recruitment strategies helped to foster welcoming feelings and widened participation in the study. For example, one participant stated: “*Yeah, and it included the entire campus regardless of your job and I think that that is a great opportunity for us all*.” Key informants also shared this view, with one stating: “*The campus was pretty energised too, and we got a good response rate because at the start of the year when they announced (the study) he said this is what I want to do and I hope everyone will jump in, and, I think we got to a lot of people and more than if you did the “hey did you want to be in that research study?*”

Theme 2—Buy-in facilitated by visible leader

This theme developed as participants discussed their perception as to what motivated them to get involved in the study. It was apparent that participants’ buy-in increased with a respected and well-liked colleague visibly leading the intervention, as suggested below: “*I think the participation and the width of the participation is much because Jonny is a well-known and well liked, well connected researcher, faculty member, member of our community(staff)*. Additionally, another participant said; “*having Jonny next door if I have been sitting about for a bit and I hear him, I’m like, you’ve been sitting for a long time?! You better stand up!*”

Theme 3—Participants curiosity and concern about their health facilitated enrolment

In this theme, participants shared that their reasons for taking part in the research was associated with interest and worry regarding their health. The theme was characterised by statements such as:
“*We’re not 30 year olds anymore most of us anyway and I think the logistics are starting to catch up you know. I’ve got friends my age who’ve had heart attacks or who are on blood pressure medicine and maybe have developed diabetes you know it’s all around me and I think that health is really really important to me.*”

#### 3.1.3. Barriers to Enrolment

Sixty-nine office-based employees who, after showing interest in the study decided not to participate, completed a separate non-participation questionnaire. Of the respondents; 18 said they were too busy, 10 felt uncomfortable with data collection, five said it was not a convenient time for them; three said they did not understand what it would entail; three said they forgot to respond; two said they were not interested in the information and one person said they already stand/move a lot in their occupation. Twenty-seven of the 69 respondents selected the “other” category. When these responses were coded, four key reasons for not participating were identified. These included; did not meet the inclusion criteria for the study (*n* = 4); medical issue, pregnancy or disability hindered participation (*n* = 11); organisational and logistical issues with recruitment (*n* = 7); perceived workload pressure (*n* = 5). One person felt they did not need the intervention.

Seven employees who participated in the baseline data collection but did not participate in the consultation completed the drop out questionnaire. Three respondents reported being “too busy” to participate. Four respondents reported scheduling/logistical issues with intervention data collection. Adding to this, one barrier to recruitment was also identified through the key informant interviews.

Theme 4—Email recruitment not suitable for all employees

It was discussed in key informant interviews that the recruitment strategy of using emails only may have been a barrier to enrolment of some types of employees. For example, one key informant stated: “*We did not successfully reach all campus employees I’d say we had under representation in the people that work in like the services like the dining services, maintenance crew, cleaners so a lot of those people also it tends to be a position where, so we recruited via email, so it assumes that people are sat at a computer so that’s not really a population that are really email users*.”

### 3.2. Effectiveness

Nine qualitative themes, and quantitative data facilitated reporting on multiple indicators of effectiveness. These indicators included; effectiveness of intervention components; additional outcomes of the intervention; and facilitators and barriers to effectiveness.

#### 3.2.1. Effectiveness of Intervention Components

Theme 1—Email intervention component was less effective

This theme developed through responses such as; “*Sorry, but I don’t really remember the weekly e-mails.*”, and “*Although I am incredibly aware of the dangers of sitting, I still do it. The emails haven’t helped me change my habits at all.*” Adding to this, data the post intervention questionnaire reflected how useful participants thought the weekly emails were; with 29.5% responding that they felt they were “*very useful*”; and 31.2% saying they were “*fairly useful*”. In addition, 19.7% felt the emails were “*somewhat useful*” and 18% felt the emails were “*minimally useful*”. One participant (1.6%) felt they were “*not at all useful*.”

Theme 2—Consultation with ActivPAL feedback was a positive experience

In this theme, participants expressed their positive opinions about the consultation, and how they perceived the consultation affected them. For example, one participant said: “*Personally, I am a visual learner so that I was shocked to see the results of how much. To see it on a graph in colour, and to see how much time I am actually sitting and because it’s 24 h. I sleep too you know! and it really captured it visually for me*.”

Additionally, questionnaire data regarding the consultation was collected through the post intervention questionnaire. In the questionnaire participants were asked how informative they felt the consultation was; with most of the participants (77.4%) feeling the consultation was either very informative (45.6%) or fairly informative (32.8%). Some participants felt the consultation was somewhat informative (19.7%), and only one participant felt the consultation was minimally informative (1.9%).

#### 3.2.2. Additional Outcomes of the Intervention

Theme 3—Intervention caused social and cultural changes which facilitated reducing sitting time

This theme developed as participants shared their perceptions of how the intervention affected the social acceptability in the office culture to stand or move, instead of sitting. For example, one participant said: “*We’ve actually, we almost are continuing the perpetuation of the (standing) culture campus wide of course doing it (standing) in the larger meetings has been, you know, everyone laughed at first, but now everyone is stood up! So it’s ok. So, I do think it’s starting a cultural shift almost like a paradigm shift to stand*”.

Theme 4—Increased education of sitting as health concern

In this theme, participants shared their belief that the intervention fundamentally changed their understanding that too much sitting is a health concern. For example, one participant said: “*I think what influenced me the most probably was the first time I sat down and received all the educational material which explained you know the benefits of standing up and the costs of staying sedentary*.” Adding to this theme, data from the post-intervention questionnaire indicated that 32.8% of participants said their awareness of sitting increased a lot, 31.1% said it increased a fair amount, 18% said it increased moderately, and 18% said it increased a little.

Theme 5—Breaks energise the brain

Participants shared how breaking up and reducing sitting time affected their energy and productivity. This is highlighted through three participants conversing about breaking up sitting time, saying: Participant A: “*Your brain is more active too I think*.” Participant B responds: “*I think you’re thinking level too*.” Participant C responds: “*Yeah and your energy*.”

Theme 6—Made changes at home to reduce sitting time.

In this theme, participants shared examples as to how the intervention affected their sitting and activity behaviours outside of the office environment. For example, one participant stated: “I *started parking my car on the other side of the parking lot when I go home, and I work from home. I have a, I guess its bar height, in my kitchen, and so instead of going into my office and working at my desk I started putting my computer and working there in the mornings*.”

#### 3.2.3. Barriers to Effectiveness

Theme 7—Concentration and focus on work tasks was a barrier to sitting

Participants expressed the difficulty they experienced trying to reduce and break up sitting time when they were concentrating and focussing on work tasks. For example, one participant stated: “*With the kind of work that we do, we are super focused when we’re working it’s almost impossible to even track the time that goes by*”. Additionally, another participant stated.” *And then I’ll get into a project and the next thing you know an hour and a half’s gone by, and I’ve not stood up at all*.”

Theme 8—Lack of management support

Participants shared their experiences of coming up against institutionalised middle management barriers to reducing and breaking up sitting time: “*It has to do with the middle manager, if you will. Who is in charge of that unit; and in some cases, you work in human resources, and you darn better be logging in and out of the computer every time you go away from your desk*.”

Theme 9—New working policy limited employees’ ability to reduce sitting time

Key informants learned through the intervention that the implementation of a new policy regarding employees working hours limited behaviour change for some employees. For example, one key informant explained: “*Those that work hourly there was a big shift this year and they had to sign in and sign out electronically so if they took a break at all like if they had to walk to their car because they forgot something, they would have to swipe out and swipe back in, and everything became very monitored they felt someone was almost breathing down their necks and that was a very common frustration. I think it rolled out right at the beginning of the study...so it was interesting to kind of see that shift of flexibility, to really not having much flexibility at all*.”

### 3.3. Implementation

Two qualitative themes, and quantitative data facilitated reporting on the cost of the intervention, and facilitators and barriers to implementation.

#### 3.3.1. Cost of Implementation

One of the lead researchers spent approximately seven hours training staff, and the consultation team spent approximately 24 h delivering the consultation to participants totalling 31 h of time spent implementing the intervention. This estimate excluded the time spent by researchers collecting data.

#### 3.3.2. Facilitators and Barriers

Results of the post intervention questionnaire indicated that 79.3% of study participants would not change anything about the intervention, while 20.7% said they would change something. Analysis of the qualitative data gave further insight into how the study was implemented, revealing two themes (one facilitator and one barrier).

Theme 1—Training procedure in place to keep consultation delivery consistent (facilitator)

Key informants explained details of how training, and the implementation of the consultation was managed. For example, one key informant explained: “*She (study researcher) actually had us come in one evening and basically kind of presented the presentation to us and had us ask questions along the way and then we had a couple days to look it over and we signed up to lead our first intervention and she sat in on it and then so she was there to chime in if we forgot something or skipped over something and afterwards she gave us some constructive feedback*.”

Theme 2—Minimal flexibility (scheduling/locations) caused fidelity issues (barrier)

This theme developed as both non-participants and intervention participants shared that the study management team were not flexible when scheduling the intervention related activities. For example, one participant stated; “*If you are going to ask for a second set of data, schedule it before the end of classes*.” Additionally, a second participant stated; “*I had emailed and said- “well could I do it another week cause it’s breaking through thanksgiving (a holiday long weekend) and I got “No this is it! If you’re gonna do it, this is the week. There aren’t any other ones (dates), they’re all signed up for*.”

### 3.4. Maintenance

Three qualitative themes facilitated reporting on facilitators and barriers to the maintenance of the intervention.

#### Facilitators and Barriers

Theme 1—Existing health and wellness programmes could facilitate maintenance (facilitator)

Both participants and key informants discussed the potential for the program to be sustained and institutionalised into the college as a part of existing workplace health and wellbeing programs. For example, one participant stated: “*I also wonder, our campus rec and our employee wellness program they do monthly wellness seminars, maybe one of them could be about sedentary behaviour and kind of just bring it to more people*.” Additionally, one key informant said: “*We already do some of those sorts of things (consultations) and I think now it might just be repurposing or reframing some of the stuff that we do the fact that some of the faculty members have done the hard work with some of the research. It’s there now. There is no reason on my end, which is the implementation and delivery of the (Health) programs, to not be able to figure that out*.”

Theme 2—Culture and philosophy of college may help facilitate long term behaviour change. (facilitator)

Participants shared how they felt the study aligned with the ethos of the college. For example, one participant said: “*I think the philosophy of spirit, mind and body that sort of puts you know, this is a healthy school we need to do this, and it just makes you think more about your body and what you’re doing and what’s good for you*.”

Theme 3—Consultation unsustainable due to the resources needed for delivery (barrier)

This theme developed as key informants described their perception that the consultation would be challenging to sustain given the resources used. For example, one key informant said: “*So a lot of what we did is not sustainable so I did it for free. I obviously wanted to do that, but it was not something I could sustain and I couldn’t do it all year long*.” Additionally, another key informant said: “*Who is doing the consultation I guess is more of the question? Just because, this has been put to me in a very informal way, and it wasn’t going to be consuming in any way, and I was just going to have my grad assistant send out an email once a month sort of thing*.”

## 4. Discussion

The RE-AIM QuEST framework was used to facilitate a mixed methods process evaluation to understand the potential for further implementation and scale-up of a consultation based intervention aimed at reducing and breaking up sitting time in the workplace. Upon interpretation of the results, there is potential for the intervention to be scaled up; however, the process evaluation reveals that there are some barriers across the RE-AIM QuEST framework which need to be addressed to improve the potential for successful scale-up. This discussion will firstly focus on the four RE-AIM dimensions assessed in this evaluation (reach, effectiveness, implementation and maintenance), and secondly give specific recommendations for scaling up the consultation based intervention; with considerations for researchers seeking to evaluate interventions for potential scalability.

### 4.1. Reach

#### 4.1.1. Facilitators

The reach of the intervention was positively affected by perceptions that “sitting” as a behaviour is easier to change than other health behaviours. Focusing an aspect of workplace health programming on reducing and breaking up sedentary behaviour may foster wider engagement in employees who are otherwise inactive. Additionally, visible leadership was important to the buy-in of participation. In this study, the intervention lead worked in the same office environment as the participants. Participants reported that this person’s presence and personal qualities were a crucial part of their motivation to be a part of the study. This aligns to other office based sitting interventions; for example, in Neuhaus et al., visible “team champions” were reported as “crucial” to the identification of behaviour change strategies suitable to the workplace, and in the promotion of the intervention to participants within the study [21].

#### 4.1.2. Barriers

Whilst over half of the employees complete the OSPASC questionnaire, the participation rate of the intervention was relatively low. Analysis of the questionnaires identified that the three most frequently reported reasons for not participating or dropping out of the intervention relate to a perceived lack of time; scheduling/logistical issues with data collection; and feeling uncomfortable with data collection. These results suggest that the perceived burden of data collection had a significant effect on participation. As an efficacy study, it is important to objectively measure both behaviour and health outcomes in early efficacy trials, therefore data collection activities may need to be adjusted to increase the participation rate and decrease the dropout rate at scale-up. There is evidence in the literature suggesting that pragmatic approaches to measurement, such as questionnaires, may need to be considered to achieve large participation rates while still measuring behavioural outcomes [22,23]. A second barrier reported identified that the email recruitment method may have missed out staff who spend a large proportion of their day sitting, but do not work on a computer (e.g., canteen staff). In a scaled-up intervention, an alternative recruitment strategy could be added, for example promotional posters to facilitate full inclusion of staff who may be highly sedentary, but are not computer-based.

### 4.2. Effectiveness

#### 4.2.1. Intervention Components

There were two intervention components; the weekly educational emails, and the consultation with Activpal feedback. Participants felt that the emails were a less effective element of the intervention. Qualitative responses suggest that the emails were predominantly used as prompts, rather than for educational content. There is evidence in the literature suggesting that prompts can be an effective intervention component to reduce and break up sitting time [24,25,26]. In Swartz et al., prompts were delivered to break-up sitting bouts once an hour via a wrist-worn device or a desktop computer application [26], and these effective methods may be better placed than emails to deliver prompts to reduce/break up sitting time.

It was clear that participants felt strongly that the consultations were informative. The qualitative responses gave specific context to this, highlighting that participants felt that the analysis of their personal Activpal data was crucial to unlocking motivation to change sitting behaviour. It is therefore clearly important that the visual feedback is taken forward in the consultation at scale-up. However, it has been suggested that the reach of the intervention was negatively affected by the participant burden created by the intensive data collection process. Additionally, resources needed to collect the data may not be pragmatic in a scaled up, more real-world intervention [27]. Therefore, if scaling up, it would be important to consider potential alternatives to giving participants behavioural feedback. Recent research suggests that mobile and wearable monitors are effective [28] and could provide feedback on sitting behaviour with minimal researcher involvement [28,29,30].

#### 4.2.2. Additional Effects on Behaviour

The results revealed several additional positive effects of the intervention, with no additional negative effects reported. In this intervention, the participants reported positive social and cultural changes towards an acceptance of reducing sitting time, and reported positive changes to sitting practices at home. Interestingly, these results contrast another study’s findings in which the office culture was identified as a barrier to reducing sitting time [31]. Additionally, a separate study demonstrated that sitting at home increased, compensating for reductions in sitting time found at work [32]. Through measuring additional behavioural outcomes, we have improved our holistic understanding of behaviour change, and the mechanisms of change. This can only help to improve future scaled up versions of the intervention by helping researchers identify where best to target efforts for improvement [10,33].

#### 4.2.3. Barriers to effectiveness

Concentration and focus on work tasks were a barrier to change in the intervention. This aligns with the results from several other studies [34,35], and suggests that office workers associate specific levels of concentration and focus needed to complete tasks, with sitting. It is unclear how this barrier could be addressed in a scaled-up intervention, and more research may be needed to understand the nature of this association. This may help identify appropriate ways to challenge normalised, habitual behaviour in the future.

The two remaining barriers (lack of management support and new working policy limited employees’ ability to reduce sitting time) that developed may relate to a common issue of “support” or “buy-in” of managers for the intervention. Minimal support was reflected in the implementation of a new workplace policy which required workers to clock in and out of working tasks on their personal computers. This affected some participants’ ability to reduce and break up their sitting time. These two barriers have the potential to significantly influence the success of a scaled-up intervention. It is therefore important that all of the gate keepers of power which could affect behaviour change are engaged in the intervention process. In a study published by Danquah et al., the research team of the scaled-up intervention introduced a buy-in scheme in which managers attended study meetings, and agreed to act as role models throughout the study [36]. This is a good example of how to ensure managers are educated and committed to employees’ engagement in health opportunities. Our evaluation highlights that if it is not carefully considered and planned for, organisational level barriers, such as mid-level policy initiatives, can significantly inhibit all other intervention components.

In office-based research we propose it may be beneficial to conceptualise the relationship of the organisational level influence on the individual and environmental levels as pictured in Figure 1. This illustrates that the organisational level (e.g., management buy-in or policy) most likely mediates/influences the response to intervention activities at both the individual, and environmental level. Furthermore, this conceptualisation more accurately illustrates that having a multi-component intervention may not elicit behaviour change if the organisational level is not carefully negotiated, to understand facilitators, and uncover barriers, to change.

### 4.3. Implementation

#### 4.3.1. Cost

Within RE-AIM, dissemination of the time and cost of the intervention [10] is considered important however the “cost/benefit” of interventions targeting office workers is seldom reported [9,37]. With limited access to information which could inform cost, the research team estimated the time needed to implement the intervention activities which equaled 31 h to deliver the consultation based intervention to 87 employees. This estimate was designed to best reflect the resources needed to scale up the already developed program, therefore it did not include the resource development time or the data collection time. This ratio of 31/87 (hours implementation/participant) has the potential to improve if the recommendations of this evaluation are implemented effectively. For most companies, time directly equates to cost; therefore, reducing the time spent implementing the intervention will be important to establishing buy-in, and ensuring uptake of workplace interventions. Reporting on time and cost could be very important to businesses which are looking to implement workplace health programs.

#### 4.3.2. Facilitators

Key informants perceived that the training procedures for the consultation were robust; explaining that each team member attended a training afternoon approximately three hours in length. They then attended a consultation delivered by the lead researcher. The lead researcher then watched an initial consultation and gave feedback at an arranged feedback meeting. Although initial resources are needed, this type of training procedure helps to ensure that deliverers are knowledgeable and that the consultations are being delivered consistently across the study. If resources become limited in a scaled-up intervention, alternative modes of delivering the training may need to be considered. For example, in Salmond and colleagues recent scaled up intervention, researchers successfully moved from in person training to online training to reduce the resources needed [38,39]. Although an investment of time would be needed initially, this method could make scaling-up training considerably more efficient.

#### 4.3.3. Barriers

Most of the participants had a positive experience with the intervention with the majority of them saying they would not change anything about the intervention. However, some employees felt that the research team could have been more flexible in the scheduling and location of intervention related visits. This may have caused some fidelity issues to the intervention protocol and should be considered prior to the scaling-up the intervention.

### 4.4. Maintenance

#### 4.4.1. Facilitators

The qualitative data revealed that both participants and key informants felt that their existing employee wellness program could facilitate maintenance of the intervention by adapting existing content and incorporating aspects of the intervention. Collaboration could reduce the resources needed to intervene, help to raise awareness as well as create motivation and buy-in that sitting is a workplace health issue of concern. When moving to scale, and implementing in a real-world office setting, there is a need to move away from singularly focused wellness interventions and towards a holistic integrated workplace wellness approach. In a resource limited environment, this will more likely elicit the motivation of both participants and employers alike, and facilitate long term maintenance of sitting interventions [40,41,42].

A second facilitator of maintenance of the program was the overarching philosophy of the college; spirit, mind and body. The philosophy directly links to health and, if used tactfully, it has the potential to make the implementation or integration of health-related interventions, programs or policy easier to justify.

#### 4.4.2. Barriers

Maintenance in RE-AIM QuEST looks at sustainability of both the intervention as a whole, and behaviour change long-term. Barriers to the sustainability of the intervention have been addressed through each of the other three sections (reach, effectiveness, implementation) in this evaluation. Measurement of behaviour change long-term is a limitation which relates to limited resources to collect objective data [8]. In the evaluation, participants shared that they have maintained behaviours to reduce and break-up sitting time, but this is not quantifiable or comparable to the baseline measurement of sitting. In a recent intervention, DeCocker and colleagues used questionnaires alongside objective measurement to easily administer and compare follow-up data to baseline [43]. DeCocker’s approach could be used to facilitate the pragmatic collection of long-term follow-up data in a scaled-up version of this intervention [43].

Key informant interviews revealed that they believed the resources allocated for data collection and delivering the consultation could not be sustained long term. This has a direct effect on the potential to scale-up the intervention in its current form. As discussed previously, there is potential to reduce the burden of data collection by moving from objective data to subjective data. Additionally, there are recent examples of scaled up interventions successfully moving from face to face delivery to virtual delivery [38,39,44,45]. For example, due to significant costs, unaffordable in the real world, Goode and colleagues transformed an effective multi-component intervention targeting sedentary office workers (BeUpStanding^TM^ program) into a web based scalable program that could be implemented directly by the workplaces [45]. Adopting this strategy may be effective in reducing the resources needed to deliver the sedentary consultation in a scaled-up intervention.

### 4.5. Considerations for Scale-Up

Recommendations for scale-up of the consultation intervention are presented in Table 4. Although specific to this intervention, these recommendations can be used to understand the modifications which may be needed to scale-up sedentary behaviour interventions in real world office settings. Engaging in this process evaluation highlights the importance of assessing additional indicators outside the effectiveness of primary outcomes before scale-up is attempted and how this might shape the modifications made to improve the likelihood of successful scaled up implementation. These results exemplify a need for a shift in approach suggested by Zamboni et al., and Reis et al.; In which researchers assess for scalability in pilot phases of research [12,46]. Put simply, if a public health problem requires wide scale implementation to have impact, then scale-up should be planned for (and evaluated) in the beginning [12,46,47,48]. This approach would mean that, rather than over resourcing pilot studies to enhance effectiveness [12] interventions are implemented with similar resources as available in the real-world settings [47]. This approach would also require an evaluation which measures additional indicators. As this evaluation has shown, measuring additional indicators will give researchers an understanding of the potential for, and modifications needed for scaled-up implementation [12,49]. This approach would not be without its challenges and may require researchers to work directly with stakeholders to co-produce, and test for, sustainable interventions under real-world conditions [40].

## 5. Strengths and Limitations

Many of the strengths of this paper are rooted in the pragmatic research approach and methodologies used. This study is one of the first to use the RE-AIM QuEST mixed methods framework to facilitate reporting on indictors which are often overlooked in research to provide detailed insight into the genuine potential for scale-up. The qualitative data was collected and analysed using up to date trustworthiness procedures and utilised epistemologically appropriate methods (e.g., TA) which aligned to the research questions. Also, through the non-participant questionnaire, the research team collected data on this challenging population to gather insight into why employees do not participate. This study is not without its limitations. Firstly, there was minimal demographic information collected about evaluation participants (e.g., ethnicity). Also, the proportion of intervention participants recruited for qualitative data collection was relatively low (15/87). Furthermore, the questionnaires used to collect data on participant/non-participant experiences were not validated. Additionally, the retrospective nature of this type of data collection could be improved upon. Future work could collect some of the implementation data prior to or during the intervention. Finally, although up to date trustworthiness procedures were used, the researchers acknowledge that inherent bias exists in qualitative data analysis, and the results should be interpreted with an understanding of this.

## 6. Conclusions

RE-AIM QuEST framework facilitated a comprehensive evaluation of the potential for further implementation and scale-up of a consultation-based workplace intervention which targeted both the reduction, and breaking up of sitting time. There is potential for the intervention to be scaled up; however, the process evaluation reveals that there are barriers across reach, effectiveness, implementation and maintenance. These barriers will need to be addressed before scale-up of the effective intervention is attempted. Recommendations for scaling up (Table 4) have been presented. Specifically, the research team should seek to use the recommendations to; reduce the participant burden of data collection and reduce the resource and cost of implementation. A shift in the approach to the research process in research fields that ultimately require scaled-up interventions to address the problem may be warranted. Interventions should assess for scalability in pilot phases of research.

## Figures and Tables

**Figure 1 ijerph-17-00239-f001:**
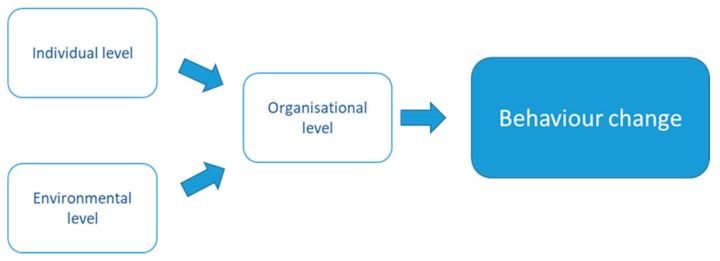
Conceptualisation of mediators of behaviour change in workplace-based research.

**Table 1 ijerph-17-00239-t001:** Example questions from focus group and interview topic guide.

RE-AIM Dimension	Questions
Reach	What convinced you to participate in the intervention? Why do you think this worked for you and not others?
Effectiveness	How did the intervention effect your sitting behaviour? Why do you think you adopted this behaviour? Were there any strategies you tried that didn’t work? Were there any barriers to you adopting new behaviours?
Implementation	Were there any challenges to being involved in the intervention? What improvements, if any, would you make to the intervention?
Maintenance	What will stop you continuing to reduce your sitting time at work? What do you think could help the intervention be maintained by the college?

**Table 2 ijerph-17-00239-t002:** Example questions from key informant interview topic guide.

RE-AIM Dimension	Questions
Reach	Were there any groups of employees you felt were not represented or missed due to the recruitment strategies undertaken? What do you think influenced the reach of the intervention?
Effectiveness	Were there any unintended or unexpected issues reported from participants?
Implementation	How did you ensure consistent implementation of the consultation? How much time was needed to train staff? Did you change or adapt the implementation as the intervention progressed?
Maintenance	What do you believe are the barriers to continuing the program? Could it become part of existing programming? If so, how?

**Table 3 ijerph-17-00239-t003:** RE-AIM dimensions, indicators assessed and the data source used to measure or inform indicators.

RE-AIM Dimension	Indicator	Measure
Reach	Quantitative How many and what proportion of the target employee population were participating in the intervention? Qualitative What were the barriers to enrolment? What explains the variation in reach, number of participants enrolled and the decline in rate of participation? What were the barriers to participation for employees? What were employees’ reasons for not participating?	Quantitative measures Participation rate = # participating/ # eligible. Drop-out rate = # signed up/ # completed assessment. Questionnaires with participants who did not take part or dropped out. Qualitative measures Interviews and focus groups with participants and key informants. Questionnaires with free text responses from both participants who dropped out and non-participants.
Effectiveness	Quantitative What were the effects of the intervention on objectively measured indicators of SB? Qualitative Were there any unintended effects of the intervention (positive or negative)? What were the conditions and mechanisms that lead to effectiveness? What adaptations are needed to improve effectiveness?	Quantitative measures Reported in Hutchison et al., 2018 [8]. Qualitative measures Interviews and focus groups with participants and key informants. Questionnaires with intervention participants.
Adoption	Not assessed	
Implementation	Quantitative What was the estimated cost of the intervention? Qualitative What were the contextual factors and processes underlying fidelity across implementation and how do we address them? What were the contextual factors and processes underlying barriers to implementation and how do we address them?	Quantitative measures # of working hours to implement intervention/ # of participants. Qualitative measures Interviews and focus groups with participants and key informants. Questionnaire responses from participants who dropped out and non-participants.

**Table 4 ijerph-17-00239-t004:** Recommendations for scaling up the sedentary behaviour intervention.

RE-AIM Dimension	Recommendation
Reach	Consider adding a non-computer-based recruitment strategy to promoting inclusion of all types of employees. Consider the addition of peer champions as visible leadership buy-in was important to initial recruitment. Reduce participant burden of outcome measurement by adjusting to be minimally intensive.
Effectiveness	Consider using the email distribution more frequently as prompts could facilitate improved effectiveness. Consider alternative ways to capture baseline sitting data for the consultation (e.g., Use data from participants existing mobile or wearable device). Adopt a system or process for buy-in of managers. Carefully consider and address organisational level barriers which could affect behaviour change at the individual and environmental level.
Implementation	Continue training procedures; however, consider alternative modes of delivering training (e.g., online training). Explore options of mobility of delivery of the intervention in convenient locations for employees (e.g., Explore if the consultation and data collection could be done in the participants’ own working environment.)
Maintenance	Work with existing workplace health program providers to explore opportunities for collaboration and integration into existing content. Utilise ethos of workplace health to increase buy-in from individual departments. Consider adding a subjective measure of behaviour change to facilitate long term follow-up data collection. Consider alternatives to the delivery of the consultation (e.g., Digital delivery) to reduce resources and cost of implementation.

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
