# Peer review of "Should We Scale-Up? A Mixed Methods Process Evaluation of an Intervention Targeting Sedentary Office Workers Using the RE-AIM QuEST Framework"

_ijerph, 2019, doi:10.3390/ijerph17010239_

Round 1

Reviewer 1 Report

The authors focused on an important topic of reducing sedentary behaviors among office workers. This is significantly associated with health improvement. The framework that the authors intended to apply was interesting. However, the paper itself lacks scientific design and in-depth data analysis.

For instance, the authors did not describe interview protocols (e.g., questions, the procedure of interviews, setup, etc.) for informants and participants; The description of quantitative data analysis was vague; No clear explanation on how SPSS was used to analyze the data, and what statistical methods were applied; No quantitative results were presented despite some simple descriptive findings; The qualitative findings lack validation; The authors may consider applying several Techniques to enhance credibility of qualitative data analysis (e.g., member checking, audit trail, triangulation).      

Author Response

Reviewer 1

We would like to thank the reviewer for their considered comments regarding our manuscript. Please find below our response to the reviewers’ comments. Changes have been highlighted in yellow in the manuscript.

Comment 1 the authors did not describe interview protocols (e.g., questions, the procedure of interviews, setup, etc.) for informants and participants;

Response: We have added further details, including two tables of example questions from the interview topic guide guides to address these issues commented on above, please see the revised manuscript. Lines 122-137.

Comment 2: The description of quantitative data analysis was vague; No clear explanation on how SPSS was used to analyze the data, and what statistical methods were applied; No quantitative results were presented despite some simple descriptive findings;

Response: We have added further details to address these issues commented on above, please see the revised manuscript. Line 213.

Comment 3: The qualitative findings lack validation; The authors may consider applying several Techniques to enhance credibility of qualitative data analysis (e.g., member checking, audit trail, triangulation)

Response: Thank you for your comment. In recent years, validation methodologies (e.g., member checking and triangulation) have been questioned in their ability to enhance credibility by several of the world experts in qualitative methodology (Braun & Clarke, 2013; Levitt et al., 2017; Smith & McGannon, 2018)- see full reference below. In light of this the authors used methods recommended to enhance trustworthiness of the data including, using the Nvivo program, creating reflexivity journals; a double coding sweep, and the “critical friend” method to interrogate the analytical decisions made during the analysis process. Additionally, we have also chosen an approach to analysis (Braun and Clark’s Thematic Analysis) which is epistemologically and ontologically flexible, aligning with the research questions of the paper. We do feel we could have given more detail in our descriptions and we feel we have addressed all of your comments in the revised manuscript. Lines 180-209.

Braun, V., & Clarke, V. (2013). Successful qualitative research: A practical guide for beginners: sage.

Levitt, H. M., Motulsky, S. L., Wertz, F. J., Morrow, S. L., & Ponterotto, J. G. (2017). Recommendations for designing and reviewing qualitative research in psychology: Promoting methodological integrity. Qualitative psychology, 4(1), 2.

Smith, B., & McGannon, K. R. (2018). Developing rigor in qualitative research: Problems and opportunities within sport and exercise psychology. International review of sport and exercise psychology, 11(1), 101-121.

Reviewer 2 Report

Thank you for the opportunity to review the manuscript titled "Should we scale-up? A mixed methods process evaluation of an intervention targeting sedentary office workers using the RE-AIM QuEST framework." This was a very interesting paper that incorporated a mixed methods approach in line with a recently published theoretical framework to promote novel practice scale up. I appreciate the authors' efforts to better understand the factors impacting the implementation of the sedentary office worker intervention. I have highlighted some major and minor comments below that I feel would strengthen the paper, especially with regard to explanation of the mixed methods data collection and coding process, as well as presentation of the results. 

Major comments:

In section 2.5, the authors indicate that they used questionnaires to assess some of the RE-AIM dimensions quantitatively. Were these questionnaires previously validated or were they developed specifically for this study? The quantitative and measures sections would benefit from substantially more detail about the types of measures used. It would be helpful to at least briefly describe each questionnaire in the text, how it was developed/any information on reliability/validity. If these questionnaires were developed solely for this project, this should be noted as a limitation as questionnaire validity is not established. Section 2.4.1 would benefit from some exemplar questions, either in table format or topics highlighted in text, to indicate the structure and topics covered in the interviews and focus groups. To what extent was there overlap in the questions asked in interviews, focus groups, and key informant interviews? In Section 2.6.1, it is unclear who completed the formal coding of the data. Was the deductive coding also completed by the lead researcher and the critical friend? Why was the critical friend selected as opposed to using objective coders? Was there a process by which the two coders established reliability? It would be helpful to have additional detail about this coding process and how reliability of coding was ensured. It would also be helpful for Section 2.6.1 to discuss how the qualitative coders differentially coded the focus group, individual interview, and questionnaire open response data. Were all transcripts and responses coded together? How were key informant interviews coded – was there a separate set of codes for these data? Finally, Section 2.6.1 should also include the analytic approach taken to determine frequently endorsed themes in the transcripts (i.e. NVivo queries?). How did the authors decide what themes were most important to report on in the results? There is clear indication for how the transcripts were coded but it is unclear how the authors obtained their results and selected exemplar quotes. I find the overall presentation of the qualitative results to be challenging to follow. Were the themes that are presented the most common themes endorsed in the qualitative interviews and focus groups? Relatedly, to make this manuscript more mixed methods oriented, it would be nice to have both the quantitative and qualitative data presented together through the entire results section. For example, the authors might consider presenting the data in line with Table 1, and present quantitative information about participation in the same section as a discussion of key themes that came up with a few exemplar quotes. Instead of listing out quotes for each theme, it would be helpful for the authors to synthesize and list the core themes from the qual and quant for each RE-AIM dimension, and then select one or two brief exemplar quotes to present in the text that best capture a few of these core themes. There are currently too many long quotes and bulleted lists in the manuscript that make it difficult to follow. The authors do not have a section to highlight the limitations of their work. I think key limitations should be noted, including potential lack of validity of some of the questionnaires used, the small sample size for focus groups and interviews, and the low representativeness of this mixed methods data collection to the full sample of 65 individuals who engaged in the intervention. The retrospective nature of this type of data collection should also be mentioned as a potential limitation, as future work could collect this type of implementation data prior to or during the intervention. Finally, it is important to address the potential bias in qualitative coding completed by the lead study researcher without a clear method to ensure reliability of coding. 

Minor comments: 

On Page 2, line 48, the authors note that existing interventions have a weakness of “failure to measure and/or report on indicators that would inform the potential for scale-up and sustainability.” It would be helpful for the authors to review the types of indicators that have an impact on intervention reach and sustainability, even if briefly. Right now it is unclear from the authors’ definitions of reach and adoption how these two domains differ as the wording of the definitions is quite similar. I would recommend noting how these two domains are different using the explicit RE-AIM QUEST definitions from Forman and colleagues (2017). I also would emphasize that reach is the number of participants who are impacted by an intervention, not necessarily those who are willing to participate. The authors do a nice job of describing the RE-AIM dimensions, it would be helpful for them to directly tie these dimensions to the RE-AIM acronym (i.e. R – reach, etc.) It would be helpful to define “intervention agents” the first time this phrase is used on page 2 line 62. I would also argue that the intervention agents are often unlikely to just be the research team, but also agencies or community partners who will be training their staff/actually using the intervention. Would recommend the authors expand their definition of intervention agents. On page 2, line 67, I would consider re-defining maintenance at the individual level. Per the Forman et al. (2017) paper, individual level maintenance involves ongoing effectiveness of the program, not simply the long-term effects of a program (which could be effective or ineffective). On page 2 line 82 there appears to be an extra question mark after “behavioural?” In the results section, approximately how many people participated in each focus group? This would be helpful to note, and perhaps also note the reason for selection of a focus group of this size in the methods section. In results, it would also be helpful for the authors to include additional demographic details. What were the racial/ethnic backgrounds of participants? What were their roles within the college and to what extent are these roles typically sedentary? “Enrollment” is spelled incorrectly in several places in the manuscript. 

Author Response

Reviewer 2

We thank the reviewer for their considered comments regarding our manuscript. Your effort in review has helped us to make significant improvements to the manuscript. Please find below our response to the reviewers’ comments. Changes are highlighted in yellow in the manuscript.

Introduction comments

Comment 1:  Page 2, line 48, the authors note that existing interventions have a weakness of “failure to measure and/or report on indicators that would inform the potential for scale-up and sustainability.” It would be helpful for the authors to review the types of indicators that have an impact on intervention reach and sustainability, even if briefly.

Response: Thank you for your comment. We have added examples to address your comment. Please see the revised manuscript. Lines 50-51

Comment 2 Right now it is unclear from the authors’ definitions of reach and adoption how these two domains differ as the wording of the definitions is quite similar. I would recommend noting how these two domains are different using the explicit RE-AIM QUEST definitions from Forman and colleagues (2017). I also would emphasize that reach is the number of participants who are impacted by an intervention, not necessarily those who are willing to participate. The authors do a nice job of describing the RE-AIM dimensions, it would be helpful for them to directly tie these dimensions to the RE-AIM acronym (i.e. R – reach, etc.)

Response:  Thank you for your comment. We agree that reach and adoption could be misunderstood, and have edited the definitions to more clearly illustrate the difference between them. We have also directly tied the dimensions to the RE-AIM acronym. Please see the revised manuscript. Lines 58-65

Comment 3: It would be helpful to define “intervention agents” the first time this phrase is used on page 2 line 62. I would also argue that the intervention agents are often unlikely to just be the research team, but also agencies or community partners who will be training their staff/actually using the intervention. Would recommend the authors expand their definition of intervention agents.

Response: Thank you for your comment.  We have addressed these issues commented on above, please see the revised manuscript. Line 64

Comment 4: Reviewer 2 - On page 2, line 67, I would consider re-defining maintenance at the individual level. Per the Forman et al. (2017) paper, individual level maintenance involves ongoing effectiveness of the program, not simply the long-term effects of a program (which could be effective or ineffective)

Response: Thank you for your comment.  We have addressed these issues commented on above, please see the revised manuscript. Line 70.

Comment 5: On page 2 line 82 there appears to be an extra question mark after “behavioural?”

Response: Thank you for your comment.  We have addressed these issues commented on above, please see the revised manuscript. Line 87

Methods comments

Comment 6: Section 2.4.1 would benefit from some exemplar questions, either in table format or topics highlighted in text, to indicate the structure and topics covered in the int

Response: Thank you for your comment. We have added further details to address these issues commented on above, please see the revised manuscript. Line 145-150. Tables 1 and 2

Comment 7: To what extent was there overlap in the questions asked in interviews, focus groups, and key informant interviews?

Response: Thank you for your comment. We have added further details to address these issues commented on above, please see the revised manuscript. Line 145-150. Tables 1 and 2: Lines 135 and 151.

Comment 8: In section 2.5, the authors indicate that they used questionnaires to assess some of the RE-AIM dimensions quantitatively. Were these questionnaires previously validated or were they developed specifically for this study? The quantitative and measures sections would benefit from substantially more detail about the types of measures used. It would be helpful to at least briefly describe each questionnaire in the text, how it was developed/any information on reliability/validity. If these questionnaires were developed solely for this project, this should be noted as a limitation as questionnaire validity is not established.

Response: Thank you for your comment. We have added further details, including a supplementary file with the questionnaires to address these issues commented on above, please see the revised manuscript. Lines 122-151

Comment 9: Reviewer 2 2.6.1 should also include the analytic approach taken to determine frequently endorsed themes in the transcripts (i.e. NVivo queries?). How did the authors decide what themes were most important to report on in the results? There is clear indication for how the transcripts were coded but it is unclear how the authors obtained their results and selected exemplar quotes. I find the overall presentation of the qualitative results to be challenging to follow. Were the themes that are presented the most common themes endorsed in the qualitative interviews and focus groups?

Response: Thank you for your comment. We do acknowledge that there are qualitative methodologies in which frequency of coding might inform theme development (E.g. Framework Analysis, Content Analysis); however, epistemologically, Braune and Clark’s Thematic Analysis (TA) does not align with these methods, and is not concerned with frequency of coding as a means of theme development. In TA the researcher is tasked with finding meaning in the data which they see as analytically important in relation to the research question. The researcher then, through the process described in the paper, pulls meanings together to identify shared meaning which they deem are analytically important to the research question. The researcher does not then pick and choose which ones to use. In the spirit of neutrality, all themes that developed should be presented, as we have done in the results section. In the section of generation of themes, the process of theme development is described in detail. 

However we acknowledge that our description could have been more detailed and we have added further details to address these issues commented on above, please see the revised manuscript. Lines 182-208

Comment 10: 2.6.1, All comments about coding it is unclear who completed the formal coding of the data.; Was the deductive coding also completed by the lead researcher and the critical friend? ;Why was the critical friend selected as opposed to using objective coders?; Was there a process by which the two coders established reliability?; It would be helpful to have additional detail about this coding process and how reliability of coding was ensured. It would also be helpful for Section 2.6.1 to discuss how the qualitative coders differentially coded the focus group, individual interview, and questionnaire open response data.; Were all transcripts and responses coded together?; How were key informant interviews coded – was there a separate set of codes for these data?

Response: Thank you for your comment. In recent years, validation methodologies (e.g., member checking and inter-rater reliability (double coding)) have been questioned in their ability to enhance credibility by several of the world experts in qualitative methodology (Braun & Clarke, 2013; Levitt, Motulsky, Wertz, Morrow, & Ponterotto, 2017; Smith & McGannon, 2018)- see full reference below. In light of this the authors used methods recommended to enhance trustworthiness (rather than improve reliability, as this is quantitative terminology which suggests certainty which does not exist as humans inherently have individual biases) of the data including, using the Nvivo program, creating reflexivity journals; a double coding sweep, and the “critical friend” method to interrogate the analytical decisions made during the analysis process.  Additionally, we have also chosen an approach to analysis (Braun and Clark’s Thematic Analysis) which is epistemologically and ontologically flexible, aligning with the research questions, and data collection methods, in this paper.

We do agree that we could have given more detail of these processes, and we feel we have addressed all of your above comments in the updated manuscript. Lines 182-208

Comment 11. The authors do not have a section to highlight the limitations of their work. I think key limitations should be noted, including potential lack of validity of some of the questionnaires used, the small sample size for focus groups and interviews, and the low representativeness of this mixed methods data collection to the full sample of 65 individuals who engaged in the intervention. The retrospective nature of this type of data collection should also be mentioned as a potential limitation, as future work could collect this type of implementation data prior to or during the intervention. Finally, it is important to address the potential bias in qualitative coding completed by the lead study researcher without a clear method to ensure reliability of coding. 

Response: We have added a limitation section and have highlighted all of the limitations you have suggested. Please see the revised manuscript. Lines 685-698

Results comments

Comment 1:  I find the overall presentation of the qualitative results to be challenging to follow. Were the themes that are presented the most common themes endorsed in the qualitative interviews and focus groups? Relatedly, to make this manuscript more mixed methods oriented, it would be nice to have both the quantitative and qualitative data presented together through the entire results section. For example, the authors might consider presenting the data in line with Table 1, and present quantitative information about participation in the same section as a discussion of key themes that came up with a few exemplar quotes. Instead of listing out quotes for each theme, it would be helpful for the authors to synthesize and list the core themes from the qual and quant for each RE-AIM dimension, and then select one or two brief exemplar quotes to present in the text that best capture a few of these core themes. There are currently too many long quotes and bulleted lists in the manuscript that make it difficult to follow.

Response: Thank you for your comments. We believe we have addressed all of the issues you have commented on above regarding the presentation of the results in the revised manuscript. Firstly, where possible, we have made sure to present quantitative and qualitative data together (E.g. Effectiveness themes 1,2 & 4) Additionally, we have also stated at the beginning of each RE-AIM sections when mix methods data is used. Furthermore, we have cut down all quotes as much as possible and we have re-structured the sections as much as possible to eliminate the fragmentation caused by bullet points. Line 219-485.   

Please note that two of the questions in this comment (“common endorsed themes and choosing only key themes) we have addressed in methods comment 9.

Comment 2 In results, it would also be helpful for the authors to include additional demographic details. What were the racial/ethnic backgrounds of participants? What were their roles within the college and to what extent are these roles typically sedentary?

Response: Thank you for your comment. We have added some demographics to this section, including self-report total daily sitting time, but no other demographics were available and this is now highlighted in the limitations section. Line 686-695

Reviewer 3 Report

General comments

First of all, this is a very relevant study, as it considers the possibilities of scaling up an intervention targeting sedentary behaviour at the workplace. Several such interventions exist and have proven effective in reducing sitting time, thus the next level is to consider how to implement these successful intervention to a wider population.

Additionally, I find the mixed-method approach very suitable for this type of study. That said the manuscript needs some overall work before being ready for publication.

Point-by-point suggestions are given below, but the main points are as follows:

Language: General awareness of repetitions, unnecessary words and the use of past/present tense. Result-section: This part is very fragmented, and I lack interpretations to the citations. Citations represent the main part of this section without many comments and with a standard (and repeated) introduction. Instead I suggest a structure were the meaning f each citation is elaborated either before or after, including further details from not included citations. Examples are given below. Include a section on strengths and limitations in the discussion.

Point by point suggestions/comments:

Abstract

20: Add information on who (participants, non-participants and keyinformants). 22: simplify example: Questionnaire data was analysed in SPSS (this type of simplification could be done several other places making the text easier to read). 24: consider another word than data collection. This could be understood as the data collection for the present study. Maybe use ‘activity monitoring/activity measure’. 27: college? Consider adding information e.g. the participating college 28: hard to read sentence, maybe because of the mentioning of recommendations.

Introduction

37-39: consider revising into more direct language 57+59: consistent order of effectiveness/efficacy 61: omit ‘within the RE-AIM’ 65: Structure with : and ; to improve readability (: after level and ; after policies) 74: Past tense

Materials and methods

78: Name of the intervention? 80: How many/who are the employees in the target group 82: Remove ? in the end 90-91: Word by word repetition of line 74-76 94-96: Move ‘Non-participants’ to the beginning of the sentence b) in order to be consistent with a) and c). 120: Omit ‘further’ – or explain 119-129: Very long description. Consider shortening/removing unnecessary words. 133: It reads as if the interviews were conducted both in person and on phone – is that true. 143: What was in the questionnaires of the other groups? You only explain about the non-participants’ questionnaire 145: Where did you get data on participation rate and hourly costs? 155-175: Altogether a very extensive description, consider shortening a bit. 155: add reference to Braun and Clarkes approach – 16? 158: Add some introduction to the following explanation 159-161: fragmented, consider revising 164: Is the reference to Nvivo? 170-171: complex sentence, consider simplification

Table 1

Be consistent in the use of tense – I suggest past tense throughout Omit ‘calculate’ in the quantitative measures (reach and implementation) Omit ‘first’ before Hutchison et al Align qualitative indicators and measures under ‘effectiveness’ and ‘implementation’

Results

186: It is for sure nice that the results are presented using the RE-AIM framework, however there are many sections and headlines and it becomes a bit fragmented. 195: (and the following with the same formatting): The bullets are a bit odd. Maybe do a new section with : after the headline instead. 197+209+220 etc. (all introduction): the wording is very similar and general (‘this theme developed…’). Avoid repetitions and be more specific to the themes and quotes. 207: visible leader. I thought this theme meant that the leader in the specific office was very visible, thus providing management support, but form the quote I get that it is the leader of the intervention. Be sure to clarify this from the very beginning/in the heading. 235: Maybe a missing ‘when’ before ‘these’? 260: Sounds like there are several components of the intervention, but as I understood its only consultation and e-mails? 279-282: Consider putting these numbers in a table or mention only some of the categories/collapse categories. The same applies to line 305-307. 290-293: as mentioned above I miss interpretation of the quotations. I am not sure how the intervention affected the social culture, is it the kick-starter of something? Please elaborate – this applies to almost all quotes. 318-323: this is only one example, but in line 318 it says ‘examples’ consider elaborating the text with further examples and then put the quote before or after as a concrete description/example. 326-334: An example of interpretation here could be to state before or after the quote that participants forget the intervention because they are too immersed in their work (my interpretation, maybe it is not right). 352: What is ‘implementation specific policy’? 367: Two themes are mentioned, but this introduction is followed by three headlines: Facilitator, barrier and cost. 385-390: This part seems very internal and is hard to understand when you do not know the project flow and thanksgiving traditions. 393: Specify whose time this is: researchers/consultation team/managers/individual participants? 404-408: Please provide interpretation – I am not sure whether this is supportive or not. 436-437: Is this time included in the cost-calculation in section 3.3.3? 427-443: there are three quotes and only one introductory line, no interpretation. I am not sure how to use/understand these quotes.

Discussion

Overall: a bit fragmented because it follows the structure of the results section. Make sure you do not just repeat the results section.

Add a section on strengths and limitations.

461: Visible leadership – as mentioned in comment for line 207, it should be clear which leadership this concerns. In Neuhaus et al team champions are local as far as I remember, but your quote considers the leadership of the intervention? 486-489: these are many numbers in the discussion, they should be in the result-section and just referred. 490-493 and 494-499: Describe the consultation before the e-mails as this is the order in the intervention. 507-510: Describe results before literature as these are conflicting. 536/figure 1: Did you get inspiration from somewhere? It seems a bit odd that this figure/framework suddenly appears in the discussion. It is almost like a result, but of what? 547: Do you have any other literature on the cost of sitting time interventions? E.g. Gao L, Flego A, Dunstan DW, Winkler EA, Healy GN, Eakin EG, Willenberg L, Owen N, LaMontagne AD, Lal A et al: Economic evaluation of a randomized controlled trial of an intervention to reduce office workers' sitting time: the "Stand Up Victoria" trial. Scand J Work Environ Health 2018, 44(5):503-511.

571: Omit number? 604: Is ‘resources’ not a barrier? 611: Consider including a reference to Chau et al. who actually piloted a web-based version of their intervention (ref: Goode AD, Hadgraft NT, Neuhaus M, Healy GN: Perceptions of an online 'train-the-champion' approach to increase workplace movement. Health Promot Int 2018.) 616: Consider moving to previous section as it seems very separate from the rest of the section on scale up. Are these recommendations all mentioned above? 652-656 repeats almost word by word line 623-627. Avoid references in the conclusion-section.

Author Response

We would like to thank the reviewer for their considered comments regarding our manuscript. Your effort in review has helped us to make significant improvements to the manuscript. Please find below our response to the reviewers’ comments. Changes are highlighted in yellow in the manuscript.

Comment 1 Abstract line 20: Add information on who (participants, non-participants and key informants). 22: simplify example: Questionnaire data was analysed in SPSS (this type of simplification could be done several other places making the text easier to read). 24: consider another word than data collection. This could be understood as the data collection for the present study. Maybe use ‘activity monitoring/activity measure’. 27: college? Consider adding information e.g. the participating college 28: hard to read sentence, maybe because of the mentioning of recommendations.

Response:  We have addressed all these comments, please see the revised manuscript.  Lines 17-32

Comment 2: All Introduction comments: 37-39: consider revising into more direct language; line 65: Structure with : and ; to improve readability (: after level and ; after policies); line 61: omit ‘within the RE-AIM’; line  7+59: consistent order of effectiveness/efficacy; 74: Past tense

Response:  We have addressed all the comments made regarding the introduction, please see the revised manuscript. Line 39-40  Line 68 line 58+62 Line 77

Comment 3: 78: Name of the intervention?; 80: How many/who are the employees in the target group; in the end; 90-91: Word by word repetition of line; 74-76 94-96:; 94-96: Move ‘Non-participants’ to the beginning of the sentence b) in order to be consistent with a) and c); 119-129: Very long description. Consider shortening/removing unnecessary words; 120: Omit ‘further’ – or explain; 133: It reads as if the interviews were conducted both in person and on phone – is that true.

Response:  We have addressed all the minor issues commented on above, please see the revised manuscript.  line 81, line 95, line 97, line 122-132 (this section was shortened but other reviewers wanted more detail), line 141 

Comment 4: 143: What was in the questionnaires of the other groups? You only explain about the non-participants’ questionnaire

Response: We have taken on board your helpful feedback and have added more detail, including a supplementary file with all the questionnaires. Please see the revised manuscript. Lines 155-168. 

Comment 5: 145: Where did you get data on participation rate and hourly costs

Response: We have provided the extra detail in the manuscript regarding the data source used to calculate participation rate and hourly cost. Lines 171-175

Comment 6- All from table 1 (now table 3) Be consistent in the use of tense – I suggest past tense throughout Omit ‘calculate’ in the quantitative measures (reach and implementation) Omit ‘first’ before Hutchison et al Align qualitative indicators and measures under ‘effectiveness’ and ‘implementation’

Response: We have addressed all the issues commented above regarding the table, please see the revised manuscript.  Line 180

Comment 7 155-175: Altogether a very extensive description, consider shortening a bit

Response: Thank you for your comment. We have taken on board your helpful feedback and attempted to shorten the section by removing some unnecessary words, however the two other reviewers wanted more detail in this section. Lines 185-213

Comment 8 155: add reference to Braun and Clarkes approach – 16?; 58: Add some introduction to the following explanation 159-161: fragmented, consider revising; 164: Is the reference to Nvivo?; 170-171: complex sentence, consider simplification

Response: We have addressed all the minor issues commented on above, please see the revised manuscript. Nvivo (12) is the program. Not a square bracket [ ] line 185, line 187 lines 185-213

Results comments

Comment 1: 186: It is for sure nice that the results are presented using the RE-AIM framework, however there are many sections and headlines and it becomes a bit fragmented

Response: Thank you for your comment. We have made efforts to reduce the fragmentation of the result by restructuring and taking out as many bullets as possible. Streamlining quotes as much as possible. Lines 222-491.

As the framework has been developed, defined, and is most often reported within, district dimensions, we do feel it is important to the integrity of the framework, and our readers, that the data on RE-AIM outcomes be presented within the dimensions. Additionally, the framework heavily informed our qualitative inquiry and deductive analysis, therefore the results are intertwined with the framework dimensions. For example, each dimension definition guided the inductive and deductive coding process  Lines 222-491

Comment 2: the wording is very similar and general (‘this theme developed…’). Avoid repetitions and be more specific to the themes and quotes.

Response: Thank you for your comment. We have addressed all the minor issues commented on above, please see the revised manuscript.  Line 254 and line 268

Comment 3 195: (and the following with the same formatting): The bullets are a bit odd. Maybe do a new section with: after the headline instead. 197+209+220 etc. (all introduction

Response: Bullets were only used because they are the next layer of sectioning of the journal manuscript template, but we agree that they did add to the fragmentation of the result section. To better streamline the results, we have cut down all quotes as much as possible and we have re-structured the sections as much as possible to eliminate the fragmentation caused by bullet points. We have eliminated as many bullets as possible. Lines 219-485

Comment 4: All comments about interpretation of quotes/themes visible leader. I thought this theme meant that the leader in the specific office was very visible, thus providing management support, but form the quote I get that it is the leader of the intervention. Be sure to clarify this from the very beginning/in the heading;

as mentioned above I miss interpretation of the quotations. I am not sure how the intervention affected the social culture, is it the kick-starter of something? Please elaborate – this applies to almost all quotes. 318-323:

 line 318 this is only one example, but in line 318 it says ‘examples’ consider elaborating the text with further examples and then put the quote before or after as a concrete description/example.

326-334: An example of interpretation here could be to state before or after the quote that participants forget the intervention because they are too immersed in their work (my interpretation, maybe it is not right)

404-408:  Please provide interpretation – I am not sure whether this is supportive or not

 427-443: there are three quotes and only one introductory line, no interpretation. I am not sure how to use/understand these quotes

Response: In order to provider clarity on the themes that have developed in the analysis we have revised each theme label and provided increased description within each theme label to more accurately reflect the central organising concept the themes developed around. This should help to signpost readers more easily to the core shared perceptions represented within the themes.

We do feel that any more interpretation would compromise the neutrality needed in a results section. As with quantitative results, qualitative results should be presented as neutral as possible so the data (quotes/themes) can be judged without influence or researcher interpretation. Also, this allows the researcher to illustrate that they have engaged in the reflexive process, which is integral to thematic analysis, and qualitative inquiry generally. As with all forms of research, the discussion section is where the researchers can start to interpret what the data actually means in relation to the research question; backing up interpretation with relevant research. 

Comment 5 235: Maybe a missing ‘when’ before ‘these’?; 260: Sounds like there are several components of the intervention, but as I understood its only consultation and e-mails?

Response: We have addressed these two minor issues commented on above, please see the revised manuscript. Line 325

Comment 6: 279-282: Consider putting these numbers in a table or mention only some of the categories/collapse categories The same applies to line 305-307. 290-293:

Response: Thank you for your comment. We feel we need to mention all of the elements of each category to be transparent in reporting the results. After having tables included in to report this information in an earlier draft, it was agreed that the tables interrupted the reading of the results section. We do feel that adding them in again may fragment the results further. 

Comment 7 67: Two themes are mentioned, but this introduction is followed by three headlines: Facilitator, barrier and cost. 385-390:

Response: I have made an amendment and labeled all themes. Lines 416- 445

Comment 8 85-390: This part seems very internal and is hard to understand when you do not know the project flow and thanksgiving traditions

Response: We changed the theme name to be more descriptive as mentioned above and have indicated that thanksgiving is a holiday. Line 443

Comment 9 393: Specify whose time this is: researchers/consultation team/managers/individual participants? 436-437: Is this time included in the cost-calculation in section 3.3.3?

Response: We have addressed these two minor issues commented on above, please see the revised manuscript. Lines 171-175 line 415

Discussion comments

Comment 1: Overall: a bit fragmented because it follows the structure of the results section. Make sure you do not just repeat the results section

Response: We thank the reviewer for this feedback. We feel that it is critical to the understanding of the paper that each RE-AIM element is discussed within a standalone section. We believe the sectioning within the discussion will help the readers of the paper navigate to the information they want to obtain. We have tried to limit repetition.

Comment 2 Visible leadership – as mentioned in comment for line 207, it should be clear which leadership this concerns. In Neuhaus et al team champions are local as far as I remember, but your quote considers the leadership of the intervention?

Response: The leadership of the intervention were local and were also office workers from the same office environment as participants. The results and discussion have been changed to clarify this relationship. Please see the revised manuscript. Lines 505-507

Comment 3: 486-489: these are many numbers in the discussion, they should be in the result-section and just referred. 490-493 and 494-499; 507-510: Describe results before literature as these are conflicting.

Response: We have addressed these minor issues commented on above, please see the revised manuscript. Lines 512-522, line 531, lines 538-541 and lines 551-556

Comment 4: 536/figure 1: Did you get inspiration from somewhere? It seems a bit odd that this figure/framework suddenly appears in the discussion. It is almost like a result, but of what?

Response:  We thank the reviewer for this comment and appreciate how presenting a figure in a discussion section of a paper may seem unusual. However, we feel the figure clearly helps to discuss the exact findings presented in the barriers to effectiveness. The figure is a unique original conceptualization clearly illustrating that organisational barriers have the ability to block all intervention components. As described, it is being suggested that if we visualise intervention components interacting in this way we may better understand how to overcome them and design better interventions in the future.  Lines 581-589

Comment 5: 571: Omit number? 604: Is ‘resources’ not a barrier? Comment: Avoid references in the conclusion-section. 652-656 repeats almost word by word line 623-627

Response: We have addressed these minor issues commented on above, please see the revised manuscript. Line 620, line 642 and lines 709-711

Comment 6: 616: Consider moving to previous section as it seems very separate from the rest of the section on scale up. Are these recommendations all mentioned above?

Response: We do feel this section discusses the broader considerations for scale-up, therefore does not fit within any of the above RE-AIM sections. All of the recommendations are mentioned in previous sections. Line 663

Comment 7: 547: Do you have any other literature on the cost of sitting time interventions? E.g. Gao L, Flego A, Dunstan DW, Winkler EA, Healy GN, Eakin EG, Willenberg L, Owen N, LaMontagne AD, Lal A et al: Economic evaluation of a randomized controlled trial of an intervention to reduce office workers' sitting time: the "Stand Up Victoria" trial. Scand J Work Environ Health 2018, 44(5):503-511.

611:Consider including a reference to Chau et al. who actually piloted a web-based version of their intervention (ref: Goode AD, Hadgraft NT, Neuhaus M, Healy GN: Perceptions of an online 'train-the-champion' approach to increase workplace movement. Health Promot Int 2018.) 61

Response: Thank you for signposting us to these two key references. They have both been used and added to the discussion. Line 657 and line 595

Round 2

Reviewer 2 Report

Thank you for the opportunity to review the revised version of this manuscript. I appreciate the authors' careful review of the comments and appropriate edits to the manuscript based on reviewer feedback. Overall, the authors have done an excellent job revising the manuscript, especially with respect to improving the presentation of the mixed method results. I have highlighted only a few minor comments below that I believe would further strengthen the manuscript prior to publication.

Page 7, line 208 - In the Generation of Themes section, it would be helpful to identify how many meetings were held with the lead researcher and critical friend to enhance reproducibility of these methods by other researchers.

Page 7, line 235-237 - I am not sure I understand this sentence "Four qualitative themes, and quantitative data highlighted facilitators and barriers to high participation which will be outlined below." Would benefit from rewording.

Results section - If possible, if might be helpful to select one longer and one shorter quote for each result or verbally report themes instead of using multiple exemplar quotes. Some of the sections (e.g. 4.4.1) have multiple long quotes that result in a lot of very brief paragraphs introducing each quote.

Watch for formatting issues throughout the paper - there are some inconsistencies in indentation of headings in the results section.

Author Response

Reviewer 2

Thank you very much for a further considered review of our manuscript. Again, you have helped us to make further improvements to the paper and you’re the time you have taken to do this is very much appreciated. Please find below our response to the reviewers’ comments.

Comment 1 Page 7, line 208 - In the Generation of Themes section, it would be helpful to identify how many meetings were held with the lead researcher and critical friend to enhance reproducibility of these methods by other researchers.

Response: Thank you for your feedback. We have addressed this comment and added the specific information regarding the frequency of meetings. Please see the revised manuscript. Line 208

Comment 2 Page 7, line 235-237 - I am not sure I understand this sentence "Four qualitative themes, and quantitative data highlighted facilitators and barriers to high participation which will be outlined below." Would benefit from rewording.

Response: Thank you for your feedback. We have addressed this comment and reworded this section to better communicate the meaning. Please see the revised manuscript. Line 235-237

Comment 3 Results section - If possible, if might be helpful to select one longer and one shorter quote for each result or verbally report themes instead of using multiple exemplar quotes. Some of the sections (e.g. 4.4.1) have multiple long quotes that result in a lot of very brief paragraphs introducing each quote.

Response: Thank you for your helpful feedback. We do feel that in the last review we made efforts to reduce the quotes within the results section and this would feel counter productive to add more quotes using one short and one long quote. In section 4.4.1 the two quotes are given because this theme developed from both intervention participants and key informants expressing very similar opinions therefore, we do feel that it is important to show an example from the two unique participation groups to be fully transparent about where the data from the theme came from.

We have picked the quotes based on representation of meaning and not length and we feel that employing a blanket rule (one short/one long) may reduce the readers understanding of the meaning of the theme. We also do not feel that we can verbally report a theme without at least one quote as we feel it would not be transparent reporting of the qualitative data and may leave the reader wondering “What did the people actually say?”

Also, some of the extra small paragraphs at the start of sections were added based on the other reviewers comments.

However, you have rightly pointed out that there are too many brief paragraphs, and further improvements to shortening quotes, and combining paragraphs would significantly help to make the results less fragmented. Where possible we have tried to reduce longer quotes and we have combined paragraphs to shorten and further improve the results section. Please see sections 3.1.2 line 254- 259; 3.2.1 line 312-318; 3.3.2 line352-356; 3.3.2; line 383- 387; 4.4.1; line 452-463; 

Comment 4- Watch for formatting issues throughout the paper - there are some inconsistencies in indentation of headings in the results section

Response: Thank you for your feedback. We have addressed this comment and all themes are now underlined and indented. Please see the revised manuscript Line 232-487

Reviewer 3 Report

The manuscript has been significantly improved and I find it almost ready for publication. 

A throughout proofreading would be perfect, as there are some minor mistakes in fonts, style and punktution e.g. in the results-section: Themes are underlines, exect in the first section (3.1.2).

Author Response

Thank you for your positive comments regarding our manuscript. Please see the below response.

Comment 1: The manuscript has been significantly improved, and I find it almost ready for publication.

A throughout proofreading would be perfect, as there are some minor mistakes in fonts, style and punktution e.g. in the results-section: Themes are underlines, exect in the first section (3.1.2)

Response: Thank you for your feedback. We have addressed this comment and all themes are now underlined and indented. Please see the revised manuscript Line 232-487